# Degree of Glomerulosclerosis in Procurement Kidney Biopsies from Marginal Donor Kidneys and Their Implications in Predicting Graft Outcomes

**DOI:** 10.3390/jcm9051469

**Published:** 2020-05-14

**Authors:** Wisit Cheungpasitporn, Charat Thongprayoon, Pradeep K Vaitla, Api Chewcharat, Panupong Hansrivijit, Felicitas L. Koller, Michael A Mao, Tarun Bathini, Sohail Abdul Salim, Sreelatha Katari, Lee S Cummings, Eddie Island, Jameson Forster, Napat Leeaphorn

**Affiliations:** 1Division of Nephrology, Department of Medicine, University of Mississippi Medical Center, Jackson, MS 39216, USA; pvaitla@umc.edu (P.K.V.); sohail3553@gmail.com (S.A.S.); 2Division of Nephrology and Hypertension, Mayo Clinic, Rochester, MN 55905, USA; chewcharat.api@mayo.edu; 3Department of Internal Medicine, University of Pittsburgh Medical Center Pinnacle, Harrisburg, PA 17105, USA; hansrivijitp@upmc.edu; 4Department of Transplant and Hepatobiliary Surgery, University of Mississippi Medical Center, Jackson, MS 39216, USA; fkoller@umc.edu; 5Division of Nephrology, Division of Nephrology and Hypertension, Mayo Clinic, Jacksonville, FL 32224, USA; mao.michael@mayo.edu; 6Department of Internal Medicine, University of Arizona, Tucson, AZ 85721, USA; tarunjacobb@gmail.com; 7Renal Transplant Program, School of Medicine/Saint Luke’s Health System, University of Missouri-Kansas City, Kansas City, MO 64110, USA; skatari@saint-lukes.org (S.K.); lscummings@saint-lukes.org (L.S.C.); eisland@saint-lukes.org (E.I.); jforster@saint-lukes.org (J.F.)

**Keywords:** procurement kidney biopsy, glomerulosclerosis, kidney transplantation, transplantation, outcomes

## Abstract

**Background:** This study aimed to assess the association between the percentage of glomerulosclerosis (GS) in procurement allograft biopsies from high-risk deceased donor and graft outcomes in kidney transplant recipients. **Methods:** The UNOS database was used to identify deceased-donor kidneys with a kidney donor profile index (KDPI) score > 85% from 2005 to 2014. Deceased donor kidneys were categorized based on the percentage of GS: 0–10%, 11–20%, >20% and no biopsy performed. The outcome included death-censored graft survival, patient survival, rate of delayed graft function, and 1-year acute rejection. **Results:** Of 22,006 kidneys, 91.2% were biopsied showing 0–10% GS (58.0%), 11–20% GS (13.5%), >20% GS (19.7%); 8.8% were not biopsied. The rate of kidney discard was 48.5%; 33.6% in 0–10% GS, 68.9% in 11–20% GS, and 77.4% in >20% GS. 49.8% of kidneys were discarded in those that were not biopsied. Death-censored graft survival at 5 years was 75.8% for 0–10% GS, 70.9% for >10% GS, and 74.8% for the no biopsy group. Among kidneys with >10% GS, there was no significant difference in death-censored graft survival between 11–20% GS and >20% GS. Recipients with >10% GS had an increased risk of graft failure (HR = 1.27, *p* < 0.001), compared with 0–10% GS. There was no significant difference in patient survival, acute rejection at 1-year, and delayed graft function between 0% and 10% GS and >10% GS. **Conclusion:** In >85% KDPI kidneys, our study suggested that discard rates increased with higher percentages of GS, and GS >10% is an independent prognostic factor for graft failure. Due to organ shortage, future studies are needed to identify strategies to use these marginal kidneys safely and improve outcomes.

## 1. Introduction

In the United States, more than 90,000 patients with end-stage kidney disease (ESKD) are currently waiting for a kidney transplant [1,2]. A significant gap between the number of kidney transplant candidates and donors remains an ongoing problem, resulting in a median wait time exceeding four years [3,4,5,6]. This delay has a dramatic impact on ESKD patients on the transplant waiting list, as their survival is, on average, below 40% after 5 years on dialysis [7]. Despite the severe organ shortage, a significant number of procured organs are discarded every year [8,9].

The shortage of deceased donor organs continues to be a problem in kidney transplantation despite the implementation of expanded criteria donor (ECD) programs in 2002 to increase the use of organs from donors with ≥60 years or comorbidities [10]. In 2013, the United Network of Organ Sharing (UNOS) Kidney Transplantation Committee approved a new allocation policy based on the kidney donor profile index (KDPI), a percentile score that compares an organ to previously recovered kidneys and signifies donor factors affecting transplant function [11]. KDPI >85% kidneys, previously designated as expanded criteria donor (ECD) kidneys, are offered to patients who have consented to accept a non-ideal renal allograft, thereby increasing access to earlier kidney transplantation [11]. Unfortunately, the discard rate for KDPI >85% kidneys continues to be high, close to 50% under the new kidney allocation system (KAS). The major determinants of discarded kidneys are donor comorbidities and procurement wedge biopsy findings, especially the percentage of glomerulosclerosis (GS) [8,12,13,14,15,16].

Despite conflicting evidence regarding the prognostic capability of histologic findings for differentiating donor kidneys at greater risk of inferior outcomes [17,18,19], the use of procurement biopsies has become an increasingly common practice, particularly in KDPI >85% kidneys in which 95% of recovered kidneys were biopsied [9,18,20]. The percentage of GS is commonly the primary biopsy information reviewed because it provides a convenient cutoff for offer turndowns [8,21,22]. This is in spite of studies noting that the percentage of GS has failed to consistently predict graft outcomes [18,21,23,24,25,26,27,28,29]. 

The aim of this study is to explore the association between the percentage of GS and graft outcomes in kidney transplant recipients who received KDPI >85% kidneys between 1 January 2005 to 2 December 2014 using the Organ Procurement and Transplantation Network (OPTN)/UNOS database.

## 2. Methods

### 2.1. Data Source and Study Population

We used the OPTN/UNOS database to identify deceased-donor kidneys recovered from January 1, 2005 to December 2, 2014 (before implementation of the kidney allocation system). The study was exempt from the institutional review board due to the publicly available nature of the de-identified database of the OPTN/UNOS database. All data used in the analysis were provided by UNOS through the Standard Transplant Analysis and Research (STAR) database. The database is a de-identified, patient-level data source that contains donor, waitlist, and transplant recipient variables derived from UNet forms for any transplant in the United States after October 1, 1987. KDPI (reference year of 2017) was calculated based on donor factors to summarize the likelihood of graft failure after deceased donor kidney transplant. Higher KDPI scores are associated with shorter estimated graft function. Although the KDPI was not formally introduced into allocation policy until implementation of the new kidney allocation system (KAS) on December 2014, the OPTN/UNOS database has KDPI values for 99% of all deceased donor recipients who underwent kidney transplantation during the study period. To assess the predictive value of procurement biopsy GS percentage in high-risk deceased donors, we only included deceased-donor kidneys with a KDPI score > 85%. We excluded recovered kidneys for dual-kidney transplant and kidneys from donors with body weight < 20 kg. Subsequently, we assess the post-transplant outcomes based on GS percentage in deceased-donor kidney transplant recipients who received kidney with KDPI > 85%. We excluded patients undergoing kidney re-transplants or multi-organ transplant from the analysis. 

### 2.2. Outcomes 

We categorized deceased donor kidneys into four groups based on the percentage of GS: 0–10%, 11–20%, >20% and no biopsy performed. We investigated the kidney discard rates and post-transplant deceased donor allograft outcomes based on GS groups. The primary outcome was death-censored graft survival. Death-censored graft survival began at kidney transplant, was followed until graft failure, defined as the requirement of renal replacement therapy and/or kidney re-transplant, and was censored at death or the end of study (6 September 2018), whichever was earlier. The secondary outcomes were patient survival, rate of delayed graft function, and 1-year acute rejection. Delayed graft function was defined as a requirement of dialysis within the first week of transplantation. As there was no statistical difference in any post-transplant outcomes between 11–20% and >20% GS (Appendix A), we combined these two groups together (>10% GS) when assessing post-transplant outcomes.

### 2.3. Covariates

Donor-related factors included donor age, sex, race, diabetes mellitus, hypertension, body mass index, the last serum creatinine before kidney procurement, donation after cardiac death, hepatitis C virus (HCV) antibody status, cause of death, and machine perfusion. Recipient-related factors included recipient age, sex, race, body mass index, diabetes mellitus, preemptive transplant, dialysis duration, and panel reactive antibody. Transplant-related factors included HLA-DR mismatch, cold ischemic time, transplant period, and induction therapy.

### 2.4. Statistical Analysis

Baseline characteristics were described using mean ± standard deviation (SD) for continuous variables or frequencies with percentage for categorical variables. Continuous variables were compared between GS groups using the student’s *t*-test or ANOVA, as appropriate. Categorical variables were compared between GS groups using the Chi-squared test. Patient survival and death-censored graft survival outcomes were estimated using the Kaplan–Meier method with significance tested using the log-rank test. The associations of the GS percentage group with death-censored graft failure and patient mortality was assessed using Cox proportional hazards analysis. The proportional hazards assumption was tested using Schoenfeld residuals (*p* = 0.29). Because the OPTN/UNOS database did not specify the date of occurrence, the associations of the GS percentage group with delayed graft function and 1-year acute rejection were assessed using logistic regression analysis. Multivariable analysis was performed to adjust for covariates associated with outcomes of interest with *p* < 0.05 in univariate analysis. All *p*-values were two-tailed, and *p*-values of <0.05 were considered significant. Stata version 13 (StataCorp, College Station, TX, USA) was used for all statistical analyses.

## 3. Results

### 3.1. Kidney Procurement Cohort and Rate of Kidney Discard

During the study period, 25,154 kidneys were recovered from deceased donors with KDPI > 85%. A total of 3014 kidneys recovered for dual-kidney transplant and 134 kidneys from donors with a body weight < 20 kg were excluded. A total of 22,006 kidneys with KDPI > 85% were included in the kidney procurement cohort. Of these kidneys, 58.0% had 0–10% GS, 13.5% had 11–20% GS, 19.7% had >20% GS, and 8.8% had no kidney biopsy performed (Appendix A). Overall, the rate of kidney discard was 48.5%; 33.6% in 0–10% GS, 68.9% in 11–20% GS, and 77.4% in >20% GS. 49.8% kidneys were discarded in the no kidney biopsy group. 

### 3.2. Kidney Transplant Recipient Cohort 

In this cohort of 22,006 deceased donor kidneys with KDPI > 85%, 10,662 kidneys were discarded. After excluding 1032 recipients with prior kidney transplants or undergoing multi-organ transplant, a total of 10,312 recipients with donor KDPI > 85% were included in the post-transplant outcome analysis. Of these patients, 75.6% had 0–10% GS, 11.9% had 11–20% GS, 4.9% had >20% GS, and 7.6% had no kidney biopsy performed (Appendix A). The median (IQR) number of glomeruli in each kidney biopsy was 47 (IQR: 28, 69). There was no association between KDPI and percent of GS (*p* = 0.70). The donor, recipient, and transplant-related characteristics stratified by percent of GS are shown in Table 1.

### 3.3. Baseline Characteristics Based on Percentages of Glomerulosclerosis

Table 2 summarizes and compares donor, recipient, and transplant-related characteristics between 0–10% and >10% GS allograft groups. Kidneys donors with >10% GS had a higher prevalence of female sex, diabetes and hypertension. Kidney donors with 0–10% GS had a higher prevalence of donation after cardiac death and positive hepatitis C antibody. Recipients of kidneys with GS > 10% were older and had longer dialysis vintage, whereas recipients of kidneys with 0–10% GS had higher panel reactive antibodies. Kidney transplants with >10% GS had more HLA DR mismatch, cold ischemic time, and thymoglobulin induction. Kidney transplants with 0–10% GS had more transplants without induction therapy. 

### 3.4. Post-Transplant Outcomes Based on Percentages of Glomerulosclerosis

The median (IQR) follow-up was 4.87 (2.90, 7.02) years after kidney transplant. During follow-up, 3015 (29.2%) patients had allograft failure, and 4433 (43.0%) patients died. A total of 1443 (14.0%) patients had acute rejection within one year, and 3436 (33.3%) patients had delayed graft function. Figure 1 compares death-censored graft survival between 0–10% and >10% GS. Graft survival rate at 5 years was 75.8% for 0–10% GS and 70.9% for >10% GS (*p* < 0.001).

In unadjusted analysis, kidneys with >10% GS were associated with a 24% higher risk of graft failure compared to kidneys with 0–10% GS (HR 1.24; 95% CI 1.13–1.36, *p* < 0.001). After adjusting for baseline donor, recipient, and transplant-related factors, kidneys with >10% GS remained significantly associated with a 27% higher risk of graft failure compared to kidneys with 0–10% GS (HR 1.27; 95% CI 1.15–1.40, *p* < 0.001) (Appendix A). Of note, there was no difference in death-censored graft survival between 11–20% GS and >20% GS (Figure 2 and Appendix A). There was no significant difference in patient survival (HR 1.03; 95% CI 0.96–1.12, *p* = 0.40), rate of acute rejection at 1-year (HR 1.13; 95% CI 0.97–1.31, *p* = 0.11), and rate of delayed graft function (HR 1.10; 95% CI 0.98–1.23, *p* = 0.11) between 0–10% GS and >10% GS (Appendix A). 

We examined the graft outcomes of >85% KDPI kidney with a low degree of GS, compared with 71–85% KDPI kidneys. The death-censored graft survival at 5 years in >85% KDPI kidneys with 0–10% GS was inferior to in 71–85% KPDI kidneys (75.8% vs. 81.2%; *p* < 0.001), as shown in Figure 3.

### 3.5. Characteristics and Outcomes of Kidneys with No Biopsy Performed

Kidneys donors with no biopsy performed were younger, more were female, and had a greater prevalence of positive hepatitis C antibody, but had a lower prevalence of diabetes, hypertension, body weight, donation after cardiac death, use of machine perfusion, and expanded criteria donation when compared with kidneys with 0–10% GS (Table 1). Graft survival rate at 5 years was comparable between 0% and 10% GS and the no biopsy group (75.8% *vs.* 74.8%; *p* = 0.62), as shown in Figure 2.

## 4. Discussion

Over 700,000 patients in the United States have ESKD, with the United States having the second-highest incidence rate of treated ESKD in the world [30]. Despite an improvement in dialysis care over the last 15 years, the overall survival on dialysis remains dismal with 22% at one year, 43% at three years and 58% at five years [31]. The risk of death is reduced by up to 66% with kidney transplantation [31]. A major limitation to increasing the number of kidney transplantations is the number of donors. It is thus of paramount importance to decrease the discard rates of high KDPI kidneys, which is estimated to be as high as 50% [8,32].

Our study showed that procurement biopsies are becoming increasingly common in marginal deceased donors in the United States. Ninety-one percent of KDPI >85% kidneys were biopsied on procurement during this study period compared to 85% between 2000 and 2003 in the United States [33]. The utility of procurement biopsies has been debated as they can delay decisions, require high resources, prolong duration of cold ischemia, and lead to unnecessary kidney discard [21,22]. Furthermore, the reliability of GS degree in predicting graft outcomes has been questioned [8,32]. While several studies have reported increased delayed graft function risk, leading to poor outcomes in kidneys with GS > 20% [19,25,28,34,35], and other studies have conversely reported similar prognoses in kidneys with GS > 20% compared to kidneys with lower GS [24,26,27,29]. Banff guidelines for procurement biopsies therefore discourage the use of rigidly defined histologic cutoffs for organ decision and allocation [19].

Using the UNOS database, we demonstrated that GS > 10% is an independent prognostic factor for graft failure in >85% KDPI kidneys, with an adjusted 1.28-fold increased risk of graft failure at 5 years when compared to kidneys with 0–10% GS. The findings of our study suggest that the use of GS percentage in procurement biopsy of >85% KDPI kidneys may improve risk stratification for recipient allograft survival. While GS > 10% was associated with a higher risk of graft failure in >85% KDPI kidneys, we did not find a difference in death-censored graft survival between allografts with 11–20% GS and >20% GS. This may suggest that >10% GS in procurement biopsies can potentially be utilized as a cutoff for risk prediction in clinical practice. Given that the presence of GS > 10% in >85% KDPI kidneys had no significant impact on delayed graft function rate, acute rejection, or patient survival, the underlying explanation for higher graft failure in GS > 10% kidneys is likely due to the progressive kidney aging process in a kidney with less residual function. As the phenotype of GS is associated with podocyte detachment and a reduced number of functioning and viable glomeruli, this leads to increasing ESKD prevalence [36,37]. It has been estimated that an allocation strategy based on pretransplant donor biopsy would increase the incidence of marginal KDPI (80% to 100%) renal transplants by over 20%, which would translate into an overall increase of 4% for the entire pool of donors [38]. Our study supports the clinical utility of the pretransplant biopsy.

This data should not discourage the use of >85% KDPI kidneys with >10% GS. There is an organ shortage with a growing number of individuals who develop ESKD every year [39] combined with a non-proportional limited supply of potential donors [32]. Overall, one-year post-transplant outcomes have improved since 2007, when the Centers for Medicare and Medicaid Services (CMS) solid organ transplant regulation was first implemented [40]. However, there is still a lack of long term graft and survival outcomes [41,42,43,44]. Although transplantation with KDPI > 85% kidneys might be associated with an increased delayed graft function rate and reduced graft survival [45], it is clearly evident based on the lower mortality rate that recipients benefit from transplantation of high-KDPI kidneys when compared with those who wait for low-KDPI kidneys [46,47]. Thus, instead of discarding >85% KDPI kidneys with >10% GS due to a higher risk of allograft loss, future studies are needed to identify techniques and strategies to improve the use and outcome of these “marginal” transplantable kidneys safely. Certain strategies are already being used, such as dual transplantation (both kidneys from one donor into the same recipient) [38,48,49,50,51,52] or creation of a protocol designed to timely identify and match suitable patient characteristics with these “marginal” kidneys (e.g., balancing the number of viable nephrons supplied within the graft versus the metabolic demand of the recipient [32]). For example, a >85% KDPI kidney with >10% GS recovered from a female donor with a low BMI may not be the best option for a male candidate with a BMI>35 kg/m^2^ [53]; further studies are needed to identify other patient and donor characteristics that would yield optimal outcomes.

Although our study aimed to assess the impact of GS degree on >85% KDPI graft outcomes, the findings of our study cannot be generalized to lower KDPI kidneys. We did compare graft outcomes between KDPI >85% kidneys with 0–10% GS to the overall KDPI 71–85% kidneys. This demonstrated that graft outcomes of KDPI >85% kidneys with 0–10% GS were inferior to KDPI 71–85% kidneys, suggesting a stronger impact of KDPI-related factors on graft outcomes over the percentage of GS on procurement biopsies. As KDPI is comprised of several clinically important donor characteristics that impact outcomes [11], it is hypothesized that these characteristics would similarly have an impact on biopsy pathology that is not limited to GS. Thus, GS percentage should not be used in isolation from other biopsy findings for individualized organ acceptance decisions. In addition, the impact GS on graft outcomes of lower KDPI scores remains unclear, since many lower KDPI kidneys are not biopsied [17,18,19].

Although our study using the UNOS database is among the largest cohorts investigating procurement biopsies with KDPI > 85%, there are some major limitations. First, there is a lack of uniform criteria for procuring, processing and interpreting procurement graft biopsies [19,54]. Core needle biopsies (during reperfusion) are usually superior to wedge biopsies (during procurement), as wedge biopsies primarily obtain sub capsular tissue, which can overestimate the amount of GS [24,26,32]. Specimens are frozen sections as opposed to paraffin-embedded tissue obtained for regular kidney biopsies or biopsies at reperfusion [21,32]. Procurement biopsies are also often interpreted by on-call general pathologists rather than nephro-pathologists. Unfortunately, the numbers of glomeruli in samples or type of pathologist were not reported in the registry. Thus, more studies aimed at optimizing assessment of procurement biopsy samples to optimally allocate organs are needed. Second, data on other important biopsy parameters in the UNOS database, such as interstitial fibrosis, tubular atrophy, and arteriosclerosis, were limited. Only 30 patients in our cohort had available reports on the degree of interstitial fibrosis, tubular atrophy, or arteriosclerosis. Therefore, future studies evaluating the predictive role of a complete histological evaluation [55], including glomerular, tubular, interstitial, and vascular compartments of >85% KDPI kidneys, are required. Furthermore, GS percentage was reported in the database as 0–10%, 11–20%, and >20%. Thus, kidney transplant outcomes using a higher cut-off of GS percentage could not be evaluated and required future studies. Furthermore, given the differences between procurement biopsies and reperfusion biopsies [18], the findings of our study cannot be generalized to reperfusion biopsies for >85% KDPI kidneys. Finally, the registry may be subjected to selection bias. Kidneys from donors that did not undergo biopsy tended to have less unfavorable clinical characteristics, than those with biopsy as demonstrated in our study (kidney donors in the no biopsy group were younger and had a lower prevalence of diabetes and hypertension), and thus had comparable graft survival rate when compared to the 0–10% GS group, but superior to the >10% GS group. Kidneys with a higher degree of GS were likely to be more carefully selected for unreported factors, including other biopsy characteristics. Alternatively, the kidney discards in each GS percentage cohort may have been impacted by other non-reported factors that influenced study outcomes.

In conclusion, we demonstrated that procurement biopsies for >85% KDPI kidneys are very commonly obtained in the United States, at a rate of 91.2%. A higher percentage of GS in >85% KDPI kidney biopsies are associated with an increased discard rate. Among KDPI >85% kidneys, GS >10% is an independent risk factor for allograft failure. However, graft survival from 0–10% GS kidneys is still inferior to kidneys with KDPI 71–85%, suggesting a stronger impact of KDPI on graft outcomes. Instead of discarding kidneys, future studies are needed to identify strategies to optimally utilize these “marginal” kidneys safely.

## Figures and Tables

**Figure 1 jcm-09-01469-f001:**
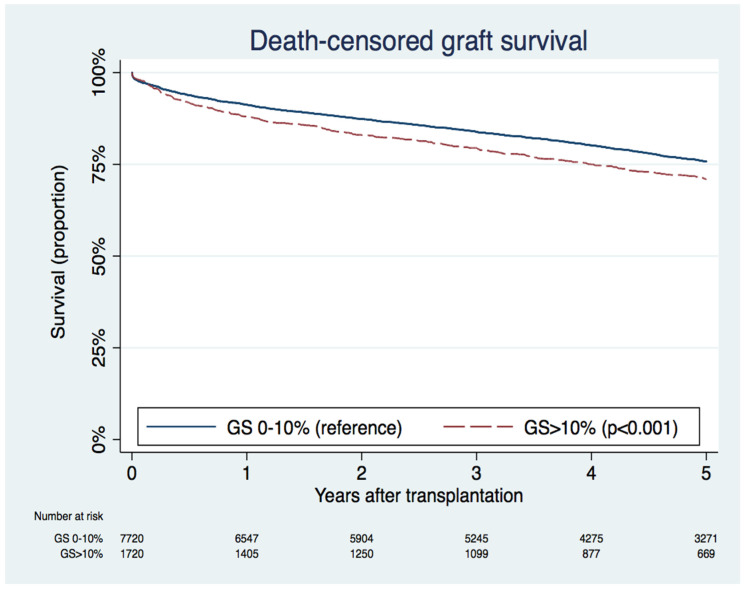
Kaplan–Meier death-censored graft survival curves between 0–10% and >10% allograft glomerulosclerosis (GS) groups.

**Figure 2 jcm-09-01469-f002:**
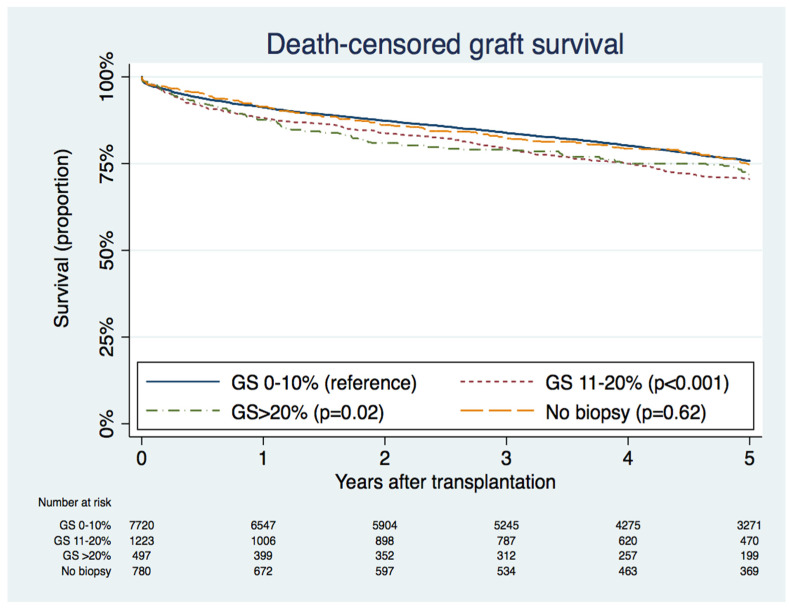
Kaplan–Meier death-censored graft survival curves according to percent glomerulosclerosis (GS) in allografts.

**Figure 3 jcm-09-01469-f003:**
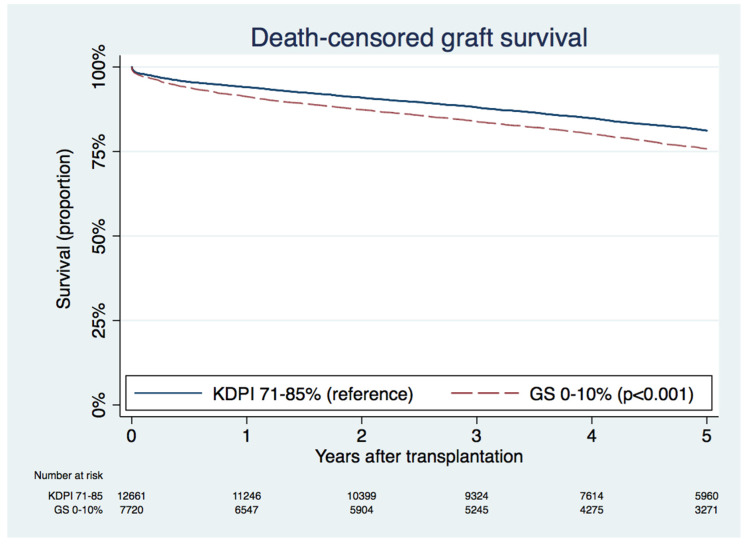
Kaplan–Meier death-censored graft survival curves between the kidney donor profile index (KDPI) 71–85% group and the KDPI >85% with 0–10% percent glomerulosclerosis (GS) group.

**Table 1 jcm-09-01469-t001:** Characteristics of donors, recipients, and transplant according to percent GS in transplanted allograft.

	Glomerulosclerosis				
	0–10%	11–20%	>20%	No-Biopsy	*p*-Value
*N*	7796	1230	500	786	
Donor					
Age (years)	60.7 ± 7.1	60.5 ± 6.9	60.6 ± 6.7	58.5 ± 7.4	<0.001
Male (%)	46.7	41.1	46.4	36.6	<0.001
Black (%)	28.8	29.5	25.2	30.0	0.26
Diabetes (%)	26.5	34.2	30.6	20.2	<0.001
Hypertension (%)	79.0	81.8	80.8	73.8	<0.001
BMI (kg/m^2^)	28.8 ± 6.9	29.2 ± 7.5	29.2 ± 7.5	27.3 ± 6.5	<0.001
Creatinine (mg/dL) before kidney procurement	1.2 ± 1.0	1.2 ± 0.6	1.2 ± 0.5	1.2 ± 1.0	0.04
Donor after cardiac death (%)	10.0	7.1	6.8	4.6	<0.001
HCV antibody positive (%)	5.8	2.4	2.8	12.9	<0.001
Cause of death (%)					
Cerebrovascular accident	78.1	79.4	83.0	82.8	0.002
Machine perfusion (%)	49.9	51.2	47.8	19.5	<0.001
Expanded criteria donor (%)	85.4	87.7	86.4	75.5	<0.001
Recipient					
Age (years)	61.5 ± 9.8	62.4 ± 9.6	61.6 ± 9.8	59.9 ± 10.6	0.001
Male (%)	64.0	63.1	63.2	63.9	0.92
Black (%)	36.4	36.8	40.6	33.3	0.07
BMI	27.9 ± 5.3	28.4 ± 5.5	27.5 ± 5.0	27.6 ± 5.4	0.001
Diabetes (%)	47.2	46.8	48.8	47.1	0.90
Dialysis duration (%)					
Preemptive	9.8	8.9	9.8	9.0	0.70
<1 years	8.7	8.1	7.0	9.5	0.41
1–3 years	29.6	26.7	30.8	30.7	0.13
>3 years	49.4	54.0	48.4	46.8	0.006
PRA (%)					
<10	81.7	84.7	81.8	78.1	0.003
10–60	12.0	9.8	12.6	15.0	0.005
>60	5.9	4.6	4.4	5.7	0.21
Missing	0.5	0.9	1.2	1.2	0.01
Transplant					
HLA DR mismatch (%)					
0	8.2	7.2	7.8	8.9	0.42
1	39.0	36.3	38.4	36.9	0.27
2	52.8	56.5	53.8	54.2	0.12
Cold ischemic time (hours)	19.5 ± 9.4	20.5 ± 9.5	19.8 ± 9.1	15.7 ± 9.2	<0.001
Transplant period					
2005–2007	28.6	22.7	35.0	45.2	<0.001
2008–2010	33.8	31.6	35.6	27.4	0.001
2011–2014	37.6	45.7	29.4	27.5	<0.001
Induction therapy (%)					
Thymoglobulin	46.1	52.3	51.4	47.3	<0.001
Alemtuzumab	14.7	13.5	16.0	9.5	0.001
Basiliximab	18.9	19.2	17.2	24.7	0.001
Other induction	7.6	8.1	5.2	7.1	0.19
No induction	16.3	11.0	13.4	15.0	<0.001

GS, glomerulosclerosis; HCV, hepatitis C virus; BMI, body mass index; PRA, panel reactive antibody; HLA, human leukocyte antigen.

**Table 2 jcm-09-01469-t002:** Comparison of donors, recipients, and transplant characteristics between GS 0–10% and GS > 10% transplanted allografts.

	Glomerulosclerosis		
	0–10%	>10%	*p*-Value
*N*	7796	1730	
Donor			
Age (years)	60.7 ± 7.1	60.5 ± 6.8	0.18
Male (%)	46.7	42.7	0.002
Black (%)	28.8	28.3	0.66
Diabetes (%)	26.5	33.1	<0.001
Hypertension (%)	79.0	81.5	0.02
BMI (kg/m^2^)	28.8 ± 6.9	29.2 ± 7.5	0.04
Creatinine (mg/dL) before kidney procurement	1.2 ± 1.0	1.2 ± 0.6	0.85
Donor after cardiac death (%)	10.0	7.0	<0.001
HCV antibody positive (%)	5.8	2.5	<0.001
Cause of death (%)			
Cerebrovascular accident	78.1	80.4	0.04
Machine perfusion (%)	49.9	50.2	0.82
Expanded criteria donor (%)	85.4	87.3	0.04
Recipient			
Age (years)	61.5 ± 9.8	62.2 ± 9.6	0.003
Male (%)	64.0	63.1	0.48
Black (%)	36.4	37.9	0.26
BMI	27.9 ± 5.3	28.1 ± 5.4	0.35
Diabetes (%)	47.2	47.4	0.86
Dialysis duration (%)			
Preemptive	9.8	9.1	0.40
<1 years	8.7	7.8	0.25
1–3 years	29.6	27.9	0.16
>3 years	49.4	52.4	0.02
PRA (%)			
<10	81.7	83.9	0.03
10–60	12.0	10.6	0.10
>60	5.9	4.6	0.03
Missing	0.5	1.0	0.01
Transplant			
HLA DR mismatch (%)			
0	8.2	7.4	0.30
1	39.0	36.9	0.10
2	52.8	55.7	0.03
Cold ischemic time (hours)	19.5 ± 9.4	20.3 ± 9.4	<0.001
Transplant period			
2005–2007	28.6	26.2	0.05
2008–2010	33.8	32.8	0.40
2011–2014	37.6	41.0	0.008
Induction therapy (%)			
Thymoglobulin	46.1	52.0	<0.001
Alemtuzumab	14.7	14.2	0.59
Basiliximab	18.9	18.6	0.79
Other induction	7.6	7.3	0.67
No induction	16.3	11.7	<0.001

GS, glomerulosclerosis; HCV, hepatitis C virus; BMI, body mass index; PRA, panel reactive antibody; HLA, human leukocyte antigen.

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
