# Peer review of "Degree of Glomerulosclerosis in Procurement Kidney Biopsies from Marginal Donor Kidneys and Their Implications in Predicting Graft Outcomes"

_jcm, 2020, doi:10.3390/jcm9051469_

Round 1

Reviewer 1 Report

This is a solid manuscript using retrospective UNOS data to determine the impact of GS on kidney transplant outcomes in the high KDPI donors.

The authors present the appropriate background, explain the methods well, and list both the classic and more subtle shortcomings of such a study.  

I would have really like to have seen an attempt to Correlate KPDI with GS.  Do the KDPI 90 kidneys have more GS than the KPDI 85%?

I think it is striking that the GS >10% was only 5% worse at 5 years.  The graft loss seems to happen early in the Kaplan Meier curves.  It might be worth pointing out that CMS focuses on the shorter-term outcomes of 1 year instead of 5 graft survival.  Maybe CMS should focus on longer-term outcomes.  This may be something that could be concretely done to discourage disincentives.

Style points

I think the percentages of the total should be included in this sentence:

"During follow-up, 3,015 patients had allograft failure, and 4,433 patients died. 1,443 patients had acute rejection within one year, and 3,436 patients had delayed graft function."

Author Response

Response to Reviewer#1

This is a solid manuscript using retrospective UNOS data to determine the impact of GS on kidney transplant outcomes in the high KDPI donors.

The authors present the appropriate background, explain the methods well, and list both the classic and more subtle shortcomings of such a study. 

Response: We thank you for reviewing our manuscript and for your critical evaluation.

Comment #1

I would have really like to have seen an attempt to Correlate KPDI with GS.  Do the KDPI 90 kidneys have more GS than the KPDI 85%?

Response:  We analyzed the correlation between KDPI (continuous variable) and percent of GS (categorical variable), as shown in below table.

0-10%

11-20%

>20%

No biopsy

p-trend

KDPI

Mean (SD)

92.38 (0.04)

92.47 (0.04)

92.19 (0.04)

91.35 (0.04)

0.70

The following statements have been added to the result as suggested.

“There was no association between KDPI and percent of GS (p=0.70).”

Comment #2

I think it is striking that the GS >10% was only 5% worse at 5 years.  The graft loss seems to happen early in the Kaplan Meier curves.  It might be worth pointing out that CMS focuses on the shorter-term outcomes of 1 year instead of 5 graft survival.  Maybe CMS should focus on longer-term outcomes.  This may be something that could be concretely done to discourage disincentives.

Response: The reviewer raises important point. We agree and thus added this point in the discussion as the reviewer’s suggestion.

“Overall, one-year post-transplant outcomes have improved since 2007, when the Centers for Medicare & Medicaid Services (CMS) solid organ transplant regulation was first implemented (40). However, there is still a lack of long term graft and survival outcomes”

Comment #3

I think the percentages of the total should be included in this sentence:

"During follow-up, 3,015 patients had allograft failure, and 4,433 patients died. 1,443 patients had acute rejection within one year, and 3,436 patients had delayed graft function."

Response: We agree with the reviewer. Percentages have been added to these statements, as suggested

We greatly appreciated the editor and reviewer’s time and comments to improve our manuscript.

Reviewer 2 Report

This work is very important in deceased donor kidney transplantation because kidney transplantation is mainly recognized as a QOL improvement therapy rather than life saving therapy. It is worthwhile to show GS is an independent risk factor of kidney graft failure when using KDPI >85% marginal donors despite of some limitations the authors mentioned.

Also, this paper showed the utility of KDPI for the field of kidney transplantation.

Author Response

This work is very important in deceased donor kidney transplantation because kidney transplantation is mainly recognized as a QOL improvement therapy rather than life saving therapy. It is worthwhile to show GS is an independent risk factor of kidney graft failure when using KDPI >85% marginal donors despite of some limitations the authors mentioned.

Also, this paper showed the utility of KDPI for the field of kidney transplantation.

Response: We thank you for reviewing our manuscript and for your critical evaluation. We appreciate the reviewer's kind comments and we agree that this study will provide important findings to the field of kidney transplantation.

We greatly appreciated the editor and reviewer’s time and comments to improve our manuscript.

Reviewer 3 Report

Cheungpasitporn et al. present a lovely manuscript on the degree of glomerulosclerosis (GS) in procurement renal biopsies from marginal donor kidneys and compare these findings with graft outcomes. The manuscript focuses on kidney donor profile index (KDPI) score > 85%, and then subcategorises this index by degree of GS. Their results show that GS >10% is an independent prognostic factor for graft failure.

Overall the paper is very well written and uses appropriate statistics. The results are compelling, yet the authors do not draw false conclusions, nor make overzealous recommendations. They appropriately note that compared with lower KDPI scores, GS <10% in the KDPI > 85% is still inferior.

My only recommendation would be to perhaps speculate further/provide references on degree of GS in KDPI <85% and impact on outcome. It is clear that other factors in the KDPI score overshadow GS alone, however it would be worth considering the impact GS has on lower KDPI scores and whether in general GS is lower.

Author Response

Response to Reviewer #3

Cheungpasitporn et al. present a lovely manuscript on the degree of glomerulosclerosis (GS) in procurement renal biopsies from marginal donor kidneys and compare these findings with graft outcomes. The manuscript focuses on kidney donor profile index (KDPI) score > 85%, and then subcategorises this index by degree of GS. Their results show that GS >10% is an independent prognostic factor for graft failure.

Overall the paper is very well written and uses appropriate statistics. The results are compelling, yet the authors do not draw false conclusions, nor make overzealous recommendations. They appropriately note that compared with lower KDPI scores, GS <10% in the KDPI > 85% is still inferior.

Response: We thank you for reviewing our manuscript and for your critical evaluation.

Comment #1

My only recommendation would be to perhaps speculate further/provide references on degree of GS in KDPI <85% and impact on outcome. It is clear that other factors in the KDPI score overshadow GS alone, however it would be worth considering the impact GS has on lower KDPI scores and whether in general GS is lower.

Response: The reviewer raises very important point. We agree and additionally discuss this point as the reviewer’s suggestion in discussion.

“In addition, the impact GS on graft outcomes of lower KDPI scores remains unclear, since many lower KDPI kidneys are not biopsied (17-19).”

We greatly appreciated the editor and reviewer’s time and comments to improve our manuscript.
